# Zero-shot Mixed Precision Quantization via Joint Optimization of Data Generation and Bit Allocation

## Abstract

Mixed-precision quantization (MPQ) aims to identify optimal bit-widths for layers to quantize a model. On the other hand, zero-shot quantization (ZSQ) aims to learn a quantized model from a pre-trained full-precision model in a data-free manner, which is commonly done by generating a synthetic calibration set used for quantizing the full-precision model. While it is intuitive that there exists inherent correlation between the quality of the generated calibration dataset and the bit allocation to the model's layers, all existing frameworks treat them as separate problems. This paper proposes a novel method that jointly optimizes both the calibration set and the bit-width of each layer in the context of zero-shot quantization. Specifically, we first propose a novel data optimization approach that takes into consideration the Gram-Gradient matrix constructed from the gradient vectors of calibration samples. We then propose a novel scalable quadratic optimization-based approach to identify the model's bit-widths. These proposals will then be combined into a single framework to jointly optimize both the calibration data and the bit allocation to the model's layers. Experimental results on the ImageNet dataset demonstrate the proposed method's superiority compared to current state-of-the-art techniques in ZSQ.

## 1 Introduction

The impressive performance of deep learning models across various fields and applications has spurred considerable interest in their applications on resource-constrained devices. As a result, model acceleration and memory optimization for deep neural networks (DNNs) have become an increasingly crucial problem. Among the prevalent network compression techniques such as pruning (He et al., 2017; Molchanov et al., 2019), knowledge distillation (Hinton et al., 2014; Romero et al., 2015), and quantization (Courbariaux et al., 2015; Rastegari et al., 2016), network quantization stands out as one of the most effective approaches. Network quantization aims to produce smaller models by representing full-precision parameters (32 bits) with significantly smaller bit-widths, e.g., 1, 2, or 4 bits), while still achieving performance comparable to full-precision models (Chen et al., 2021; Dong et al., 2019; Yang & Jin, 2020; Wei et al., 2022; Nagel et al., 2020).

Zero-shot quantization is a quantization approach when there is no access to any part of the original data. By leveraging information from the full-precision model, ZSQ generates a small set of synthetic data and exploits it as calibration data for the quantization process. Prominent ZSQ methods (Choi et al., 2021; Jeon et al., 2023; Cai et al., 2020; Li et al., 2023) take advantages of statistics from the batch normalization (BN) layers of the full-precision model to generate synthetic samples such that the feature distributions of generated samples match the BN statistics. Other approaches focus on boundary information, generating data near the decision boundary of the full-precision model (Choi et al., 2021; Li et al., 2023; Qian et al., 2023a).

Instead of using the same bit-width for all layers, mixed-precision quantization (MPQ) seeks to enhance performance by allocating higher bit-widths to crucial network layers. MPQ methods can be generally divided into two main categories: search-based approaches and sensitivity-based methods. Search-based approaches, such as those proposed by (Wang et al., 2018; Deng et al., 2023), employ reinforcement learning (RL) to optimize bit-widths for network layers. In contrast, sensitivity-based

methods, including (Cai et al., 2020; Dong et al., 2019; 2020; Chen et al., 2021), use layer sensitivity as a proxy measure to evaluate the contribution of each layer to the final model performance. These techniques determine sensitivity scores for layers using predefined heuristics and then formulate an integer programming problem to allocate bit-widths based on these sensitivity scores, while adhering to computational and memory constraints. However, none of the aforementioned methods consider the impact of data quality during the mixed-precision identification process.

Despite substantial research on mixed-precision quantization and zero-shot quantization, these topics are often treated as separate problems. Given that most existing mixed-precision methods employ data-based heuristics, it is intuitive to assume that the quality of the calibration set influences the bit-width selection in mixed-precision settings, and vice versa. In this paper we propose a novel approach for zero-shot mixed-precision quantization in which the data generation and bit allocation are jointly optimized, thereby enhancing the efficiency of quantized models. To our best knowledge, this paper is the first work that jointly optimize the data generation and the bit allocation in the context of zero-shot mixed-precision quantization. Furthermore, we propose a novel data generation approach that is based on the Gram-Gradient matrix of the generated calibration data. Specifically, we firstly show that the optimization of the training set is equivalent to maximizing its gradient matching with the validation set during the training process. We then theoretically show that this optimization is equivalent to matching the Gram-Gradient matrices of the two sets. In addition, we propose a scalable quadratic optimization approach to optimize the model bit-widths, that takes into account the impact of different layer bit-widths to the model's gradient.

The contributions of this work can be summarized as follows:

❶ To the best of our knowledge, this paper is the first one that proposes a mechanism to jointly optimize both the data generation and the bit allocation for zero-shot mixed-precision setting. ❷ We propose a novel approach for optimizing the calibration data based on the Gram-Gradient matrix of that calibration set. ❸ We propose a scalable quadratic optimization approach that takes into account the impact of bit-widths setting to the gradient of the model for bit allocation optimization. ❹ Experimental results demonstrate that our novel zero-shot mixed-precision quantization (ZMPQ) method outperforms the state-of-the-art ZSQ methods under low-bit quantization settings. In addition, although our proposed ZMPQ approach does not require real calibration data, our method achieves competitive results compared to mixed-precision methods that require real data for the quantization process.

## 2 RELATED WORK

### 2.1 UNIFORM QUANTIZATION

Network quantization is a family of techniques to compress network size and accelerate model inference, by representing full-precision weights with low-bit ones. Among them, uniform quantization is the most popular approach thanks to its simplicity. The de-quantized weight $\hat{w}$ of a uniformly quantized model can be determined by the quantizer $Q_b$ as:

$$\hat{w} = Q_b(w; s) = s \times \text{clip}\left(\left\lfloor \frac{w}{s} \right\rceil, n, p\right), \tag{1}$$

where $\lfloor . \rceil$ is the rounding-to-nearest function, $clip()$ is the clipping function, while $s$ represents the scaling factor. $n$ and $p$ are respectively the upper and lower clipped values, which are often set to $n = 0$ and $p = 2^b - 1$ in the unsigned $b$-bit quantized case, or $n = -2^{b-1}$ and $p = 2^{b-1} - 1$ for the signed $b$-bit case. In order to improve the efficiency of uniform quantization, there has been several prominent post training quantization (PTQ) approaches (Wei et al., 2022; Jeon et al., 2023) that use AdaRound (Nagel et al., 2020) which learns a rounding variable $v$, taking values of either 0 or 1. This approach adjusts the quantization equation to:

$$\hat{w} = s \times \text{clip}\left(\left\lfloor \frac{w}{s} \right\rfloor + v, n, p\right). \tag{2}$$

In this paper, our method will also employ the quantization mechanism of AdaRound (Nagel et al., 2020) to ensure a fair comparison.

## 2.2 ZERO-SHOT QUANTIZATION

As the solution for scenarios where the original data is not available, Zero-shot quantization (ZSQ) approaches attempt to generate synthetic data by leveraging information from the full-precision models. A conventional method for synthesizing such data is to minimize the cross-entropy loss based on the full-precision model's prediction and the synthetic label. One of the pioneering works in ZSQ, ZeroQ (Cai et al., 2020), identified that there exists distributional mismatch between the synthetic data and real data, which leads to substantial performance degradation. In order to combat this problem, ZeroQ proposes to take advantage of the batch normalization (BN) statistics derived from the full-precision model, resulting in significant performance enhancements. Subsequently, advanced ZSQ data generation techniques have emerged, taking into account this idea and additional elements for data generation. For instance, Genie (Jeon et al., 2023) proposes a framework that learns both the generator and its inputs concurrently. Qimera (Choi et al., 2021) and HAST (Li et al., 2023) propose to generate boundary-supporting samples with different approaches. While Qimera produces samples within the decision boundary between classes by combining class embeddings, HAST incentivizes samples based on their loss uncertainty. DSG (Qin et al., 2021) highlights the limited heterogeneity in synthetic samples resulting from BN statistics optimization and suggests introducing a margin threshold when minimizing BN statistics mismatch, to enhance sample variety. On the other hand, AdaDFQ (Qian et al., 2023a) seeks to optimize both boundary information and data diversity.

## 2.3 MIXED-PRECISION QUANTIZATION

Mixed-precision quantization techniques can be broadly categorized into two main categories: search-based and sensitivity-based approaches. Search-based methods, exemplified by HAQ (Wang et al., 2018) and AutoQ (Lou et al., 2019), utilize reinforcement learning for bit-width optimization of network layers. The performance of the mixed-precision quantized network serves as the reward signal, while their action space consists of all possible bit-width configurations. A notable search-based MP method recently is EMQ (Dong et al., 2023), which introduces a proxy search framework through evolving algorithms that automatically generate proxies to guide the mixed-precision process. On the other hand, sensitivity-based methods such as ZeroQ (Cai et al., 2020), HAWQ (Dong et al., 2019), HAWQ-V2 (Dong et al., 2020) and MPQCO (Chen et al., 2021) leverage layer sensitivity as a surrogate metric to estimate the impact of quantization on overall model performance. This family of technique assess layer sensitivity scores for all layer with predefined heuristics, before formulating an integer programming problem that allocates layer bit-width according to their sensitiveness, under memory and computational constraints. A recent sensitivity-based mixed-precision quantization method, CLADO (Deng et al., 2023), successfully take into account the inter-dependency between network layers when selecting their bit-width. Unfortunately, similar to conventional mixed-precision quantization techniques, CLADO (Deng et al., 2023) requires non-trivial time complexity for their bit allocation. In order to address the computational bottleneck problem, OMPQ (Ma et al., 2023), a prominent mixed precision quantization method, introduces an orthogonality metric to measure the correlation between layer orthogonality. This metric is used to determine the optimal bit-width configuration for different layers by assigning more bit to layers with stronger orthogonality. However, most of the prior mixed-precision quantization works does not consider the impact of data quality to the bit allocation algorithm.

## 3 METHOD

### 3.1 PROBLEM DEFINITION

Given a validation set $X^{(V)}$ and a deep learning model $\mathtt{f}(.)$ with pre-trained weights $\theta_{FP}$, our major objective is twofold: to *generate a synthetic dataset $X^{(T)}$* and to *identify a set of bit-widths B* for the quantized network $\theta_Q$, such that their combination results in the best performance for the lower-bit network $\mathtt{f}(\theta_Q)$ over the validation set $X^{(V)}$. In the realm of zero-shot quantization, the validation set does not exist, so we only assume it here for explanation. Additionally, the transfer of knowledge from a model $\theta_{FP}$ to a quantized model $\theta_Q$ is commonly facilitated by optimizing

layer-wise reconstruction losses, assessed on a calibrated dataset $X^{(T)}$:

$$\mathcal{L}_R(\theta_Q, \theta_{FP}, X^{(T)}, B) = \frac{1}{|X^{(T)}|} \sum_{i=1}^{|X^{(T)}|} \sum_{l=1}^{L} \|\mathtt{f}(\theta_{FP}, \mathbf{x}_i, l) - \mathtt{f}_B(\theta_Q, \mathbf{x}_i, l)\|^2, \qquad (3)$$

where $\mathtt{f}(\theta_{FP}, \mathbf{x}_i, l)$ and $\mathtt{f}_B(\theta_Q, \mathbf{x}_i, l)$ respectively denote the $l^{th}$-layer outputs of the full-precision and quantized models under the $B$ bit-width setting, given the input sample $\mathbf{x}_i$ and $|.|$ signifies the cardinality of a given set. $\theta_Q$ and $\theta_{FP}$ represent the quantized model weights (by rounding to nearest) before training and the full-precision model weights, respectively.

## 3.2 DATA OPTIMIZATION

Supposed the training set is generated by a generator $\mathcal{G}(.)$ and a set of embedding vectors $Z \coloneqq \{\mathbf{z}_i\}_{i=1}^{|X^{(T)}|}$, i.e. $X^{(T)} \coloneqq \{\mathbf{x}_i^{(T)} | \mathbf{x}_i^{(T)} = \mathcal{G}(\mathbf{z}_i)\}$, we want to enhance the model's performance on the validation set $X^{(V)}$ after the calibrate it using the generated data $X^{(T)}$. Our optimization objective for the generated dataset is:

$$X^{(T)} = \arg \min_{X^{(T)}} \mathcal{L}_R(\theta_Q^*, \theta_{FP}, X^{(V)}, B))$$

$$s.t. : \theta_Q^* = \arg \min_{\theta_Q} \mathcal{L}_R(\theta_Q, \theta_{FP}, X^{(T)}, B), \qquad (4)$$

where $\theta_Q^*$ is the model weights after updating $\theta_Q$ with $X^{(T)}$ under the bit-width setting $B$. Define $\delta_{\theta_Q} = \theta_Q^* - \theta_Q$ as the weight difference of the quantized model $\theta_Q$ before and after quantized with the training dataset $X^{(T)}$. In practice, we only approximate the calibrated model using only one step gradient descent, which yields $\delta_{\theta_Q} = -\alpha \nabla_{\theta_Q} \mathcal{L}_R(\theta_Q, \theta_{FP}, X^{(T)}, B)$, where $\alpha$ denotes the learning rate. Using the first order Taylor expansion for the reconstruction loss $\mathcal{L}_R(\theta_Q^*, \theta_{FP}, X^{(T)}, B)$ at $\theta_Q$ we have:

$$\arg \min_{X^{(T)}} \mathcal{L}_R(\theta_Q^*, \theta_{FP}, X^{(V)}, B) = \arg \min_{X^{(T)}} \underbrace{\mathcal{L}_R(\theta_Q, \theta_{FP}, X^{(V)}, B)}_{\text{independent from } X^{(T)}} + \nabla_{\theta_Q} \mathcal{L}_R(\theta_Q, \theta_{FP}, X^{(V)}, B)^T (\theta_Q^* - \theta_Q)$$

$$= \arg \min_{X^{(T)}} \nabla_{\theta_Q} \mathcal{L}_R(\theta_Q, \theta_{FP}, X^{(V)}, B)^T (\theta_Q^* - \theta_Q)$$

$$(5)$$

Denoting $\nabla_{\theta_Q} \mathcal{L}_R(\theta_Q, \theta_{FP}, X^{(T)}, B) = \mathcal{J}_T^{(\theta_Q)}$, $\nabla_{\theta_Q} \mathcal{L}_R(\theta_Q, \theta_{FP}, X^{(V)}, B) = \mathcal{J}_V^{(\theta_Q)}$ and replacing $(\theta_Q^* - \theta_Q)$ by $\delta_{\theta_Q}$ we have:

$$\arg \min_{X^{(T)}} \mathcal{L}_R(\theta_Q^*, \theta_{FP}, X^{(V)}, B) = \arg \min_{X^{(T)}} -\alpha \mathcal{J}_V^{(\theta_Q)^T} \mathcal{J}_T^{(\theta_Q)}$$

$$= \arg \max_{X^{(T)}} \frac{1}{|X^{(V)}|} \sum_{i=1}^{|X^{(V)}|} \frac{1}{|X^{(T)}|} \sum_{j=1}^{|X^{(T)}|} \mathcal{J}_{V,i}^{(\theta_Q)^T} \mathcal{J}_{T,j}^{(\theta_Q)},$$

$$= \arg \max_{X^{(T)}} \frac{1}{|X^{(V)}|} \sum_{i=1}^{|X^{(V)}|} \frac{1}{|X^{(T)}|} \sum_{j=1}^{|X^{(T)}|} I(x_i^{(V)}, x_j^{(T)}),$$

$$(6)$$

where $\mathcal{J}_{T,j}^{(\theta_Q)}$ and $\mathcal{J}_{V,i}^{(\theta_Q)}$ denote the gradient vectors of reconstruction loss evaluated on the $j^{th}$ sample in the training set $X^{(T)}$ and the $i^{th}$ sample in the validation set $X^{(V)}$, w.r.t. the model weight $\theta_Q$. The term $I(x_i^{(V)}, x_j^{(T)}) = \mathcal{J}_{V,i}^{(\theta_Q)^T} \mathcal{J}_{T,j}^{(\theta_Q)}$ is a gradient matching score, denoting how well the model can learn sample $x_i^{(V)}$ indirectly through training sample $x_j^{(T)}$. Ideally, we want the gradient matching score of all validation samples $x_i^{(V)}$ to remain high throughout the entire training process. Since the gradients of samples change over time, simply assigning $x_j^{(T)} = \arg \max \sum_{i=1}^{|X^{(V)}|} I(x_i^{(V)}, x_j^{(T)}) \quad \forall j \in 1, 2, ..., |X^{(T)}|$ will lead to a suboptimal solution. Intuitively, if the gradient matching between two samples is high enough, they will share similar

characteristics, making their gradient matching score more robust over time. Therefore, we modify the above term to reward couples of samples with high gradient matching score:

$$X^{(T)} = \arg\max_{X^{(T)}} \frac{1}{|X^{(V)}|} \sum_{i=1}^{|X^{(V)}|} \sqrt[k]{\frac{1}{|X^{(T)}|} \sum_j I(x_i^{(V)}, x_j^{(T)})^k}$$

$$= \arg\max_{X^{(T)}} \frac{1}{|X^{(V)}|} \sum_{i=1}^{|X^{(V)}|} I(x_i^{(V)}, X^{(T)}), \tag{7}$$

where $I(x_i^{(V)}, X^{(T)}) = \sqrt[k]{\frac{1}{|X^{(T)}|} \sum_j I(x_i^{(V)}, x_j^{(T)})^k}$ denotes the gradient matching score of the whole training set $X^{(T)}$ to validation sample $x_i^{(V)}$. The parameter $k$ can raise the attention on training sample $x_j^{(T)}$ that has high gradient matching with $x_i^{(V)}$. When $k = 1$, the objective is exactly the gradient matching term in Eq. (6). When $k$ increases, $\sqrt[k]{\frac{1}{|X^{(T)}|} \sum_j I(x_i^{(V)}, x_j^{(T)})^k} \approx \sqrt[k]{\frac{1}{|X^{(T)}|} max_j I(x_i^{(V)}, x_j^{(T)})^k} \sim max_j I(x_i^{(V)}, x_j^{(T)})$ will encourage each validation sample $x_i^{(V)}$ to have high gradient matching with at least one training sample $x_j^{(T)}$. The optimal $X^{(T)}$ occurs when $X^{(T)} = X^{(V)}$, implying $I(x_i^{(V)}, X^{(T)}) = I(x_i^{(V)}, X^{(V)}) \quad \forall i = 1, 2, \ldots, |X^{(V)}|$. When that happens, we call the set $X^{(T)}$ is $k$-equivalent to $X^{(V)}$. We can optimize this objective, by matching the Gram-Gradient matrix of the two sets, according to Definition 1 and Theorem 3.1, using the loss in Eq. (8):

$$\mathcal{L}_{GRAM}(X^{(T)}, X^{(V)}) = log(1 + e^{\eta(1 - cos(G^k(X^{(T)}), G^k(X^{(V)})))}), \tag{8}$$

where $cos(G^k(X^{(T)}), G^k(X^{(V)}))$ denotes the cosine similarity between the two vectors obtained by flattening the Gram-Gradient matrices $G^k(X^{(T)})$ and $G^k(X^{(V)})$; $\eta$ is a hyper-parameter. Note that in the context of post-training quantization, each block of the network is usually quantized sequentially, so the final reconstruction loss is only used to update the last block. Therefore, our Gram-Gradient matrix only requires the gradient w.r.t. the weights of the last block.

**Definition 1.** *For any vector $a = \{a_i\}_{i=1}^{|\theta_Q|}$, the $k^{th}$ order Gram matrix $G^k$ of this vector is a $k^{th}$ order tensor in which the size of each dimension is $|\theta_Q|$, defined as:*

$$G_{t_1, t_2, \ldots, t_k}^k(a) = a_{t_1} a_{t_2} \cdots a_{t_k} \text{ for } 1 \leq t_1, t_2, \ldots, t_k \leq |\theta_Q| \tag{9}$$

*$\{t_i\}_{i=1}^k$ is any groups of indices. Given the training set $X^{(T)} = \{x_i\}_{i=1}^N$ with the corresponding set of gradient vectors $\{\mathcal{J}_{T,i}^{\theta_Q}\}_{i=1}^N$, each with length $|\theta_Q|$, the Gram-Gradient matrix order $k$ of this dataset $X^{(T)}$ can be defined as:*

$$G^k(X^{(T)}) = \frac{1}{|X^{(T)}|} \sum_{i=1}^{|X^{(T)}|} G^k(\mathcal{J}_{T,i}) \tag{10}$$

**Theorem 3.1.** *Two sets $X^{(T)}$ and $X^{(V)}$ are $k$-equivalent when the $k^{th}$ order Gram-Gradient matrix of $X^{(T)}$ matches the corresponding of $X^{(V)}$.*

Please see Section A.1 in the Appendix for the proof of our Theorem 3.1.

In practice, as we found the magnitude of the gradient vectors vary a lot more than their direction during the quantization process, therefore to make a stable training, we normalize the gradient vectors before constructing the Gram-Gradient matrix. Additionally, we also want the feature distributions of generated samples match the feature distributions of original data. Therefore, we encourage $X^{(T)}$ to have similar batch normalization (BN) statistics stored in the BN layers of the full-precision model $\theta_{FP}$, by introducing the BN loss $L_{BN}$:

$$\mathcal{L}_{BN}(\theta_{FP}, X^{(T)}) = \sum_{j=1}^L (||\mu_j^{(s)} - \mu_j||^2 + ||\sigma_j^{(s)} - \sigma_j||^2), \tag{11}$$

where $\mu_j^{(s)}$ and $\mu_j$ are respectively the mean output values of the synthetic dataset $X^{(T)}$ from the full-precision model at the $j^{th}$ layer and the BN statistic of the full-precision model from the same layer, while $\sigma_j^{(s)}$ and $\sigma_j$ are the corresponding standard deviations.

At the start of the training, we try to match the distribution of the training set with the real dataset, by warming up the training generator $\mathcal{G}(.)$ and the set of training embedding vectors $Z$ using the BN loss $\mathcal{L}_{BN}$. After that, we samples neighbor samples of the warm-up set as the validation set for the Gram-Gradient loss $\mathcal{L}_{GRAM}$, by mixing the warm-up embeddings $X^{(V)} := \{\mathbf{x}_i^{(V)} | \mathbf{x}_i^{(V)} = \mathcal{G}(\beta \mathbf{z}_i + (1 - \beta)\mathbf{z}_j)\} \quad \forall\, 0 \le i, j \le |X^{(T)}|, \beta \in (0, 1)$, as these samples are likely to share similar feature distribution with the real data if the warm-up generator $\mathcal{G}(.)$ and embedding vectors $Z$ are trained well enough. The final optimization objective for $X^{(T)}$ after the warm-up stage is:

$$\mathcal{L}_{FINAL}(\theta_{FP}, X^{(T)}) = \mathcal{L}_{BN}(\theta_{FP}, X^{(T)}) + \lambda \mathcal{L}_{GRAM}(X^{(T)}, X^{(V)}). \tag{12}$$

### 3.3 BIT-WIDTH OPTIMIZATION

When we assign a full-precision model to a set of bit-widths, we are adding quantization noise to each layer of the network according to its given bit-width. Our objective is to identify the set of bit-widths that can minimize the final loss of the model (after rounding to the nearest value using that bit-width set) over the hidden validation set $X^{(V)}$.

Because our framework is a joint optimization approach, when optimizing the bit-width of the model, we assume that our training set has been trained sufficiently well, such that minimizing the calibration loss on the training set $X^{(T)}$ is equivalent to minimizing it on the validation set $X^{(V)}$. Let $\mathbf{C}$ represent all possible bit choices for each layer, e.g., $\mathbf{C} = \{2, 4, 8\}$. We define $B$ as a binary vector of size $|\mathbf{C}|L$, representing the concatenation of all $\mathbf{C}$-length bit-choice one-hot encodings from each layer of the network, where $B_{(i-1)|\mathbf{C}|:i|\mathbf{C}|}$ is the one-hot encoded bit selection for the $i^{th}$ layer. Here, $L$ is the number of layers. The final loss of the quantized model is defined as:

$$B = \arg\min_{B} \quad \mathcal{L}_R(\theta_Q, \theta_{FP}, X^{(T)}, B)$$

$$= \arg\min_{B} \quad \underbrace{\mathcal{L}_R(\theta_Q, \theta_{FP}, X^{(T)}, B) - \mathcal{L}_R(\theta_{FP}, \theta_{FP}, X^{(T)}, B)}_{\text{Second-order Taylor for } \mathcal{L}_R(\theta_Q, \theta_{FP}, X^{(T)}, B) \text{ at } \theta_{FP}} + \underbrace{\mathcal{L}_R(\theta_{FP}, \theta_{FP}, X^{(T)}, B)}_{0}$$

$$\approx \arg\min_{B} \quad \underbrace{\nabla_{\theta_{FP}} \mathcal{L}_R(\theta_{FP}, \theta_{FP}, X^{(T)}, B)^T}_{\text{Insignificant magnitude}} (\theta_Q - \theta_{FP}) + \frac{1}{2}(\theta_Q - \theta_{FP})^T \mathcal{H}^{(\theta_{FP})}(\theta_Q - \theta_{FP})$$

$$\approx \arg\min_{B} \quad \triangle_{\theta,B}^T \mathcal{H}^{(\theta_{FP})} \triangle_{\theta,B}, \tag{13}$$

where $\triangle_{\theta,B} = \theta_Q - \theta_{FP}$ denotes the change in weights when we quantize the full-precision network weights with nearest rounding using the bit-width set $B$.

We need some preparation steps in order to transform Eq. (13) into a quadratic optimization problem w.r.t. variable $B$. Let us define a matrix $\triangle$ with size $|\theta_Q| \times |\mathbf{C}|L$, in which each column $\triangle^{(i,j)} = \triangle_{(i-1)|\mathbf{C}|+j}$ represents quantization noise added to the weights of the full-precision model, when we quantize the $i^{th}$ layer to $\mathbf{C}_j$ bits, while keeping other layers at full-precision. When we only quantize the $i^{th}$ layer to $\mathbf{C}_j$ bits, the change in weights of the whole model is $\triangle^{(i,j)} = [0, 0, \dots, \theta_{Q,j}^i - \theta_{FP}^i, \dots, 0]^T$, where $\theta_{Q,j}^i$ and $\theta_{FP}^i$ are respectively the weights of the $i^{th}$ layer after quantized to $\mathbf{C}_j$ bits and full-precision weights. Given the vector $B$ that indicates the bit allocation for all $L$ layers, the quantization noise to the whole quantized network, can be estimated as:

$$\triangle_{\theta,B} = \triangle B. \tag{14}$$

Additionally, for each $i^{th}$ layer with bit option $\mathbf{C}_j$, we apply a first-order Taylor approximation to the vector-valued function $\nabla_{\theta_{FP}} \mathcal{L}_R(\theta_{FP} + \triangle^{(i,j)}, \theta_{FP}, X^{(T)}, B) = \mathcal{J}_T^{(\theta_{FP} + \triangle^{(i,j)})}$ at $\theta_{FP}$ to have:

$$\mathcal{J}_T^{(\theta_{FP} + \triangle^{(i,j)})} = \mathcal{J}_T^{(\theta_{FP})} + \mathcal{H}^{(\theta_{FP})}(\theta_{FP} + \triangle^{(i,j)} - \theta_{FP})$$

$$\approx \mathcal{H}^{(\theta_{FP})} \triangle^{(i,j)}. \tag{15}$$

The term $\mathcal{J}_T^{(\theta_{FP})}$ is negligible, as the gradient w.r.t. the full-precision model weights is close to 0, so we can estimate $\mathcal{H}^{(\theta_{FP})} \triangle^{(i,j)}$ as the gradient of the full-precision model after adding quantization noise $\triangle^{(i,j)}$ to the model weights. Therefore, we can easily calculate the matrix $\mathcal{M} = \mathcal{H}^{(\theta_{FP})} \triangle$

---

**Algorithm 1** Joint optimization framework for zero-shot mixed precision quantization.

1: **Train**($\theta_{FP}$, $\mathcal{G}$, $N_w$, $N_d$, $N_q$).
2: $\theta_{FP}$: The full-precision model.
3: $\mathcal{G}$: The generator.
4: $N_w$: Number of warm-up iterations.
5: $N_d$: Number of iterations to optimize data each round.
6: $N_q$: Number of iterations for model quantization.
7: Initialize $\theta_Q$ from $\theta_{FP}$.
8: Initialize $\mathcal{G}$ and $Z \sim \mathcal{N}(\mathbf{0}, \mathbf{I})$.
9: Warm-up $\theta_Q$, $\mathcal{G}$, $z$ with BN loss (11) for $N_w$ iterations.
10: **while** not converged **do**
11:     Optimize and update bit-widths of all layers by solving Eq. (17)
12:     Optimize the calibration set with Eq. (12) in $N_d$ iterations
13: **end while**
14: Get the dataset $X^{(T)} := \{\mathbf{x}_i^{(T)} | \mathbf{x}_i^{(T)} = \mathcal{G}(\mathbf{z}_i)\}$.
15: **for** $t = 1$ to $N_q$ **do**
16:     $\theta_Q = \theta_Q - \alpha \frac{\partial \mathcal{L}_R(\theta_{FP}, \theta_Q, X^{(T)})}{\partial \theta_Q}$
17: **end for**
18: **return** $\theta_Q$.

---

with size $|\theta_Q| \times |\mathbf{C}|L$, each column $\mathcal{M}^{(i,j)} = \mathcal{M}_{(i-1)|\mathbf{C}|+j} = \mathcal{H}^{(\theta_{FP})}\triangle^{(i,j)} \approx \mathcal{J}_T^{(\theta_{FP}+\triangle^{(i,j)})}$. Combining this with Eq. (14) and Eq. (13), the optimization in Eq. (13) becomes:

$$B \approx \arg\min_B \quad \triangle_{\theta,B}^T \quad \mathcal{H}^{(\theta_{FP})} \quad \triangle_{\theta,B}$$

$$= \arg\min_B \quad B^T \underbrace{\triangle^T \mathcal{H}^{(\theta_{FP})}} \quad \triangle B$$

$$= \arg\min_B \quad B^T \underbrace{\mathcal{M}^T} \quad \triangle B$$

$$= \arg\min_B \quad B^T \quad \mathcal{A} \quad B. \tag{16}$$

Let $\mathbf{D} = \{|\theta_{Q,j}^i| * \mathbf{C}_j\}_{i,j}$ be a vector of length $|\mathbf{C}|L$, where each value $\mathbf{D}^{(i,j)} = \mathbf{D}_{(i-1)|\mathbf{C}|+j}$ denotes the total bits count of the $i^{th}$ layer when that layer is set to $\mathbf{C}_j$ bits, $\mathbf{B}_{budget}$ represents the model size target. Finally, we have a quadratic integer programming problem with constraints:

$$B = \arg\min_B \quad B^T \mathcal{A} B$$

$$\text{s.t: } B_i \in \{0,1\} \quad \forall 1 \le i \le |\mathbf{C}|L$$

$$\sum_i^{i+|\mathbf{C}|} B_i = 1 \quad \forall i = k|\mathbf{C}| \text{ and } 0 \le k \le L-1$$

$$B^T \mathbf{D} \le \mathbf{B}_{budget} \tag{17}$$

It is worth noting that although our final objective function (17) has similar form as CLADO (Deng et al., 2023), there are significant differences between ours and CLADO (Deng et al., 2023). Firstly, our method takes into consideration the gradient information caused by the changes in the bit-width of each layer, while CLADO (Deng et al., 2023) measure the changes in the final loss when a layer's bit-width decreases. Secondly, regarding the computational complexity, our method requires the computation of the matrix $\mathcal{M}$ which has the size $|\theta_Q| \times |\mathbf{C}|L$. For each column of $\mathcal{M}$, it requires a forward and backward through the network. Therefore, our method requires $\mathcal{O}(|\mathbf{C}|L)$ forward and backward passes through the network which is cheaper than CLADO (Deng et al., 2023) which requires $\mathcal{O}(|\mathbf{C}|^2 L^2)$ forward passes through the network.

## 3.4 FINAL ALGORITHM

Initially, the model is set to uniform bit-width, and we warm up the calibrated set $X^{(T)}$ using a data generation method. The warm-up dataset will then be used to sample validation samples $X^{(V)}$

by mixing the embedding. The framework will then alternatively optimize the model's bit-width setting with the newly updated set $X^{(T)}$ and update the synthetic set $X^{(T)}$ using the model with the updated bit-width setting, until convergence. After the joint optimization stage, we calibrate the current quantized model under new bit-width setting using the newly generated set of training data $X^{(T)}$ with the reconstruction loss in Eq. (3). Algorithm 1 presents the overall algorithm of our proposed method.

## 4 EXPERIMENTS

### 4.1 EXPERIMENTAL SETUP

**Dataset and network architectures.** We evaluate our methodology using the ImageNet (Russakovsky et al., 2015) dataset, a common benchmark in zero-shot quantization studies. We validate our approach on ResNet-18 (He et al., 2016), ResNet-50 (He et al., 2016), and MobileNetV2 (Sandler et al., 2018) architectures.

**Quantization setting.** We follow the approach outlined in Genie (Jeon et al., 2023), starting with a standard uniform quantization framework that incorporates an adaptive rounding variable, as detailed in section 2.1. The weight bit-widths of both the initial and last layers are set to 8 bits for initialization. Additionally, following BRECQ (Li et al., 2021), we set the activation bit-widths for the second and last layers to 8 bits, while other layers adopt the uniform activation bit setting. The model's bit budget, $B_{budget}$, is equal to the size of the initial model.

**Implementation details.** To warm up the synthetic dataset, we utilize a generator and 256-dimensional embedding vectors for each mini-batch of generated images using Genie model (Jeon et al., 2023). Initially, we set the learning rate for the generator to 0.1 and for the embeddings to 0.01. We set $k$ to 10, and use the $10^{th}$ order Gram-Gradient matrix for our data optimization. The parameter $\lambda$ in Eq. (12) is set to 0.02, while $\eta$ in Eq. (8) is set to 2, 1 and 1 for ResNet-18, ResNet-50 and MobileNetV2, respectively. We adopt the Adam optimizer (Kingma & Ba, 2014) for both the generator and the data embeddings, and we employ ExponentialLR and ReduceLRon-Plateau learning rate schedules for the generator and the data embeddings, respectively. Throughout all experiments, we use a batch size of 128 during data generation and a batch size of 32 during the quantization process. To demonstrate the effectiveness of our proposed method, we benchmark our model against different SOTA zero-shot quantization models on various low-bit settings. Following previous works (Jeon et al., 2023; Qian et al., 2023a) and incorporating an additional ultra-low-bit setting (2/2), we employ three distinct quantization configurations for the zero-shot ImageNet experiments: 2/2, 3/3, and 4/4 bit-widths. We then compare our proposed approach with recent leading zero-shot quantization models, i.e., Qimera (Choi et al., 2021), Genie (Jeon et al., 2023), AdaDFQ (Qian et al., 2023a), IntraQ (Zhong et al., 2022) and AdaSG (Qian et al., 2023b). For a fair comparison, we generate a total of $1,024$ images, following Genie (Jeon et al., 2023). Additionally, we also present the performance of the current leading mixed-precision quantization models (Ma et al., 2023; Dong et al., 2023; Yang & Jin, 2020; Cai et al., 2020; Li et al., 2021; Choi et al., 2018) under different model size constraints $\{4.0, 4.5, 5.81\}$.

### 4.2 COMPARISON WITH THE STATE-OF-THE ART ZERO-SHOT QUANTIZATION METHODS

Table 1 presents the comparative results of our proposed approach and other state-of-the-art zero-shot quantization methods when evaluated on the ImageNet dataset. The results of the competitors are collected from (Jeon et al., 2023). Additionally, we compare our proposed method with the combination of generated data from (Jeon et al., 2023) and our proposed bit allocation. It is clear that our proposed approach consistently outperforms competitor quantization methods across different bit-width settings and network architectures. The improvement is clearer with MobileNetV2, showing increases of 11.76%, 7.47%, and 1.69% for the */2, */3, and */4 settings, respectively, which confirms the effectiveness of our proposed approach. Furthermore, our joint optimization framework outperforms the Genie (Jeon et al., 2023) model combined with our bit allocation method in all settings, indicating the effectiveness of the joint optimization of both data generation and bit allocation for zero-shot mixed-precision quantization.

Table 1: Comparisons of Top-1 classification accuracy (%) with state-of-the-art zero-shot quantization frameworks on the ImageNet dataset. Genie + MPQ denotes the combining of generated data using Genie (Jeon et al., 2023) model with our bit allocation method. Results marked with (*) are reproduced using the official released code of the corresponding paper. Our method maintains the same model size constraint as other methods in the corresponding settings.

| Method | W/A | ResNet-18 | ResNet-50 | MobileNetV2 |
|---|---|---|---|---|
| | Full precision | 71.01 | 76.63 | 72.20 |
| Genie (Jeon et al., 2023) | 2/2 | 53.74 | 56.81 | 11.93 |
| Genie + MPQ (Ours) | */2 | 58.33 | 63.85 | 23.39 |
| Ours | */2 | **58.60** | **64.23** | **23.69** |
| Qimera (Choi et al., 2021) | 3/3 | 1.17 | - | - |
| AdaSG (Qian et al., 2023b) | 3/3 | 37.04 | 16.98 | 26.90 |
| IntraQ (Zhong et al., 2022) | 3/3 | - | - | - |
| AdaDFQ (Qian et al., 2023a) | 3/3 | 38.10 | 17.63 | 28.99 |
| Genie (Jeon et al., 2023) | 3/3 | 66.89 | 72.54 | 55.13* |
| Genie + MPQ (Ours) | */3 | 67.33 | 73.75 | 62.45 |
| Ours | */3 | **67.57** | **73.83** | **62.60** |
| Qimera (Choi et al., 2021) | 4/4 | 63.84 | 66.25 | 61.62 |
| AdaSG (Qian et al., 2023b) | 4/4 | 66.50 | 68.58 | 65.15 |
| IntraQ (Zhong et al., 2022) | 4/4 | 66.47 | - | 65.10 |
| AdaDFQ (Qian et al., 2023a) | 4/4 | 66.53 | 68.38 | 65.41 |
| Genie (Jeon et al., 2023) | 4/4 | 69.66 | 75.57* | 68.38 |
| Genie + MPQ (Ours) | */4 | 69.88 | 75.84 | 69.95 |
| Ours | */4 | **70.05** | **75.87** | **70.07** |

Table 2: The comparisons of our proposed method with state-of-the-art mixed-precision quantization methods under different model size budgets on ResNet-18. The last column **Zero-shot** denotes whether the framework requires real data.

| Method Name | W/A | Model Size (MB) | Top-1 (%) | Zero-shot |
|---|---|---|---|---|
| FracBits-PACT (Choi et al., 2018) | */* | 4.5 | 69.10 | - |
| OMPQ (Ma et al., 2023) | */4 | 4.5 | 68.89 | ✗ |
| EMQ (Dong et al., 2023) | */4 | 4.5 | **69.66** | ✗ |
| Ours | */4 | 4.5 | 69.38 | ✓ |
| ZeroQ (Cai et al., 2020) | 4/4 | 5.81 | 21.20 | ✓ |
| BRECQ (Li et al., 2021) | 4/4 | 5.81 | 69.32 | ✗ |
| PACT (Choi et al., 2018) | 4/4 | 5.81 | 69.20 | ✗ |
| HAWQ-V3 (Yao et al., 2020) | 4/4 | 5.81 | 68.45 | ✗ |
| FracBits-PACT (Choi et al., 2018) | */* | 5.81 | 69.70 | ✗ |
| OMPQ (Ma et al., 2023) | */4 | 5.5 | 69.38 | ✗ |
| EMQ (Dong et al., 2023) | */4 | 5.5 | **70.12** | ✗ |
| Ours | */4 | 5.81 | 70.06 | ✓ |
| BRECQ (Li et al., 2021) | */8 | 4.0 | 68.82 | ✗ |
| OMPQ (Ma et al., 2023) | */8 | 4.0 | 69.41 | ✗ |
| EMQ (Dong et al., 2023) | */8 | 4.0 | **69.92** | ✗ |
| Ours | */8 | 4.0 | 69.86 | ✓ |

## 4.3 COMPARISONS WITH THE STATE-OF-THE-ART MIXED-PRECISION QUANTIZATION METHODS

Table 2 shows the superiority of our method compared to other prominent mixed-precision quantization methods (Ma et al., 2023; Dong et al., 2023; Yang & Jin, 2020; Cai et al., 2020; Li et al., 2021; Choi et al., 2018). We conduct the experiments using ResNet-18 frameworks with different fixed model size budgets $4.0, 4.5$, and $5.81$. As shown in Table 2, our proposed method outperforms current state-of-the-art mixed-precision quantization methods in all settings, except for EMQ (Dong et al., 2023). It is worth noting that our method is a zero-shot mixed-precision quantization frame-

work for PTQ, while most currently available mixed-precision approaches, including EMQ (Dong et al., 2023), require real data, as denoted in the last column of Table 2.

## 4.4 ABLATION STUDIES AND VISUALIZATION

### 4.4.1 IMPACT OF EACH COMPONENT ON THE MODEL'S PERFORMANCE

Table 3: Ablation study on the impact of individual components on the method's performance. Genie (Jeon et al., 2023), the current SOTA method for zero-shot PTQ, is used as the baseline, in the first row of each benchmark. Genie consists of a data generation part and a quantization part (with uniform bit allocation). **Data** refers to replacing the data generation part of Genie with our data generation method. In the case of mixed-precision, we constrain the number of bits used so that the model size remains the same as with the uniform bit-width setting. **Mixed** denotes whether we use our mixed-precision approach.

| W/A | Model size (MB) | Data | Mixed | Top-1 (%) |
|---|---|---|---|---|
| */2 | 3.16 | | | 53.74 |
| | | ✓ | | 54.16 |
| | | | ✓ | 58.33 |
| | | ✓ | ✓ | **58.60** |
| */4 | 3.16 | | | 65.0 |
| | | ✓ | | 65.15 |
| | | | ✓ | 67.05 |
| | | ✓ | ✓ | **67.22** |
| */3 | 4.48 | | | 66.89 |
| | | ✓ | | 66.97 |
| | | | ✓ | 67.33 |
| | | ✓ | ✓ | **67.57** |
| */4 | 5.81 | | | 69.66 |
| | | ✓ | | 69.85 |
| | | | ✓ | 69.88 |
| | | ✓ | ✓ | **70.05** |

Table 3 demonstrates the impact of each component (data optimization and bit-width optimization) on the final model's performance. We either replace Genie's data generation mechanism with our approach or apply our proposed mixed-precision method using Genie's synthetic data. The results show that the bit-width optimization approach has a significantly greater impact on model performance than the data generation approach. Furthermore, jointly optimizing both data and bit-width allocation yields even greater improvements, which demonstrates the effectiveness of our joint optimization framework.

### 4.4.2 IMPACT OF HYPER-PARAMETERS TO OUR METHOD

Please see Section A.2 in the appendix for the ablation study of our hyperparameters.

### 4.4.3 VISUALIZATION

Please see Section A.4 in the appendix for examples of our synthetic Imagenet calibration images.

## 5 CONCLUSION

In this paper, we introduce a novel approach for zero-shot mixed-precision quantization. Our approach identifies the mixed-precision configurations as a scalable quadratic optimization, and theoretically formulates Gram-Gradient matrix matching for zero-shot quantization. Furthermore, we explore the relationship between data optimization and bit-width selection demonstrating their mutual enhancement within a unified optimization framework. Extensive experiments across various quantization settings and model budgets confirm the effectiveness of the proposed approach.

ACKNOWLEDGMENTS

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

## A  APPENDIX

### A.1  PROOF FOR THE THEOREM

*Proof.* From the definition we have:

$$
\begin{aligned}
I(x_i^{(V)}, x_j^{(T)})^k &= (\mathcal{J}_{V,i}^{(\theta_Q)}{}^T \mathcal{J}_{T,j}^{(\theta_Q)})^k \\
&= (\sum_{t=1}^{|\theta_Q|} \mathcal{J}_{V,i,t}^{(\theta_Q)} \mathcal{J}_{T,j,t}^{(\theta_Q)})^k \\
&= \sum_{1 \le t_1, t_2, \ldots, t_k \le |\theta_Q|} \prod_{q=1}^{k} \mathcal{J}_{V,i,t_q}^{(\theta_Q)} \prod_{q=1}^{k} \mathcal{J}_{T,j,t_q}^{(\theta_Q)} \\
&= \sum_{1 \le t_1, t_2, \ldots, t_k \le |\theta_Q|} G_{t_1,t_2,\ldots,t_k}^k(x_i^{(V)}) G_{t_1,t_2,\ldots,t_k}^k(x_j^{(T)}) \\
&= G^k(x_i^{(V)}) \circ G^k(x_j^{(T)}),
\end{aligned}
\tag{18}
$$

where $\circ$ denotes the sum of all elements after element-wise product of 2 matrices. Then we have:

$$
\begin{aligned}
I(x_i^{(V)}, X^{(T)}) &= \sqrt[k]{\frac{1}{|X^{(T)}|} \sum_j I(x_i^{(V)}, x_j^{(T)})^k} \\
&= \sqrt[k]{\frac{1}{|X^{(T)}|} \sum_j G^k(x_i^{(V)}) \circ G^k(x_j^{(T)})} \\
&= \sqrt[k]{G^k(x_i^{(V)}) \circ (\frac{1}{|X^{(T)}|} \sum_j G^k(x_j^{(T)}))} \\
&= \sqrt[k]{G^k(x_i^{(V)}) \circ G^k(X^{(T)})}
\end{aligned}
\tag{19}
$$

Therefore, if the two sets have the same Gram-Gradient matrix (i.e., $G^k(X^{(T)}) = G^k(X^{(V)})$), we have:

$$
\begin{aligned}
I(x_i^{(V)}, X^{(T)}) &= \sqrt[k]{G^k(x_i^{(V)}) \circ G^k(X^{(T)})} \\
&= \sqrt[k]{G^k(x_i^{(V)}) \circ G^k(X^{(V)})} \\
&= I(x_i^{(V)}, X^{(V)}) \quad \forall i = 1, 2, \ldots, N,
\end{aligned}
\tag{20}
$$

which means $X^{(T)}$ and $X^{(V)}$ are $k$-equivalent  $\square$

### A.2  ABLATION STUDIES ON THE HYPER-PARAMETERS

We conduct ablation studies on the impact of different hyper-parameters $(k, \eta, \lambda)$ of the frameworks, as illustrated in Tables A.1, A.2 and A.3, respectively. The experiments all hyper-parameters are performed on ResNet18 with 4/4 setting using 1024 synthetic images.

Table A.1: Change in performance w.r.t. $k$ in Eq. (12)

| $k$ | 4 | 8 | 10 | 16 | 20 |
|---|---|---|---|---|---|
| ZMPQ (Ours) | 69.88 | 69.91 | 70.05 | 69.98 | 69.95 |

Table A.2: Change in performance w.r.t. $\eta$ in Eq. (8)

| $\eta$ | 1 | 2 | 4 | 8 |
|---|---|---|---|---|
| ZMPQ (Ours) | 69.94 | 70.05 | 69.94 | 70.00 |

Table A.3: Change in performance w.r.t. $\lambda$ in Eq. (12)

| $\lambda$ | 0.01 | 0.02 | 0.05 | 0.1 |
|---|---|---|---|---|
| ZMPQ (Ours) | 69.95 | 70.05 | 69.83 | 69.88 |

Table A.4: The comparisons of our proposed approach with state-of-the-art mixed-precision quantization methods under different model size budgets (1.3 MB and 1.5 MB) on MobileNetV2. The last column **Zero-shot** denotes whether the framework requires real data.

| Method Name | W/A | Model Size (MB) | Top-1 (%) | Zero-shot |
|---|---|---|---|---|
| Baseline | 32/32 | 13.4 | 72.49 | - |
| FracBits-PACT (Choi et al., 2018) | */* | 1.3 | 68.99 | ✗ |
| OMPQ (Ma et al., 2023) | */8 | 1.3 | 69.62 | ✗ |
| EMQ (Dong et al., 2023) | */8 | 1.3 | **70.72** | ✗ |
| Ours | */8 | 1.3 | 69.70 | ✓ |
| FracBits-PACT (Choi et al., 2018) | */* | 1.84 | 69.9 | ✗ |
| OMPQ (Ma et al., 2023) | */8 | 1.5 | 70.28 | ✗ |
| EMQ (Dong et al., 2023) | */8 | 1.5 | 70.75 | ✗ |
| Ours | */8 | 1.5 | **71.27** | ✓ |

## A.3 ADDITIONAL MIXED-PRECISION QUANTIZATION RESULTS ON MOBILENETV2

Table A.4 shows the performance of our method compared to other state-of-the-art mixed-precision quantization methods using MobilenetV2, with two different model size constraints (1.3 MB and 1.5 MB). Compared to EMQ (Dong et al., 2023), the result of our proposed method is lower in the setting with a 1.3 MB model size constraint, while achieving greater performance on the 1.5 MB benchmark.

## A.4 VISUALIATION OF SYNTHETIC IMAGES

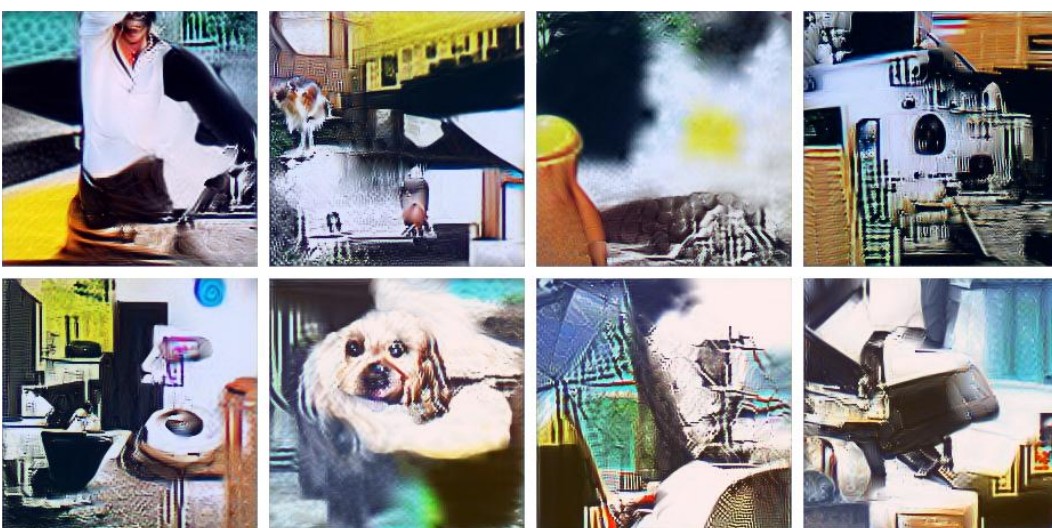

Figure A.1: The synthetic images generated by our method.

