# OpenReview forum: "Zero-shot Mixed Precision Quantization via Joint Optimization of Data Generation and Bit Allocation"
_ICLR.cc/2025/Conference — Submitted to ICLR 2025_

### Official Review · Reviewer_mxwe · 2024-10-27

**Soundness:** 3
**Presentation:** 3
**Contribution:** 2
**Rating:** 8
**Confidence:** 4

**Summary:**

They introduce a jointly optimization framework for zero-shot quantization that alternatively optimize the calibration data and model bit-widths

**Strengths:**

The authors propose a mixed precision quantization method for ZSQ from the perspective of data quality. I support this work on the good topic.

**Weaknesses:**

My main concern is that network optimization from a gradient perspective may be passive given the poor quality of the available data, and the intuition is that bypassing the gradient information and using a gradient-free search method may be a better option.

**Questions:**

1. The quantizer setup. The GDFQ/Qimera/AdaDFQ series compared in this paper all use perchannel quantizer and naive optimization by STE. However, these details are not elaborated for the proposed method.
2. The proposed method also works for conventional PTQ when the calibration data set is assumed not to fit the model statistics, which indicates that the method is not completely limited to ZSQ. Authors should provide more targeted explanations or provide experiments of the proposed method on PTQ for corroboration. Perhaps the proposed method can be used to build benchmarks on PTQ?
3. The traditional benchmark on CIFAR data set has not been adopted, and the authors are requested to add explanations.
4. line235: "When that happens, we call the set X^(T) is k-equivalent to X^(V) . We can optimize this objective, by matching the Gram-Gradient matrix of the two sets, according to Definition 1 and Theorem 3.1" ZSQ does not provide a validation set. I am trying to doubt the assumption in the article that when two samples are from synthetic datasets, their matching does not indicate the correctness of the feature?
5. The generator configuration is not well aligned with the comparison method because Genie's generator is a newer version and this part needs further ablation.
6. Lack of overhead analysis.

---

> ### Author Response · Authors · 2024-11-21
> **Response to Reviewer mxwe**
>
> # Response to Reviewer mxwe
>
> **We greatly appreciate the time and effort the Reviewer dedicated to considering our paper. Here are our responses to all concerns raised by the Reviewer.**
>
> ## Weakness
>
> **W1**:
> *My main concern is that network optimization from a gradient perspective may be passive given the poor quality of the available data, and the intuition is that bypassing the gradient information and using a gradient-free search method may be a better option.*
>
> **Answer**:
> Most existing zero-shot quantization methods rely on gradient-based optimization to generate synthetic training data with different optimization criteria (e.g., hardness [1], boundary supporting [2]). In our framework, the synthetic dataset is warmed-up by aligning its Batch Norm statistics with those of the full-precision model before the optimization process. Therefore, the dataset used for optimization likely shares similar features and distributions with real data. Experimental results on various architectures have validated the effectiveness of our proposed method. While gradient-free search could be a potential approach for ZSQ, it is not the focus of our paper.
>
> ## Question
>
> **Q1**:
> *The quantizer setup. The GDFQ/Qimera/AdaDFQ series compared in this paper all use per-channel quantizer and naive optimization by STE. However, these details are not elaborated for the proposed method.*
>
> **Answer**:
> For the model's weights quantization, similar to current state-of-the-art methods for zero-shot/mixed-precision quantization (Genie [3], BRECQ [4], EMQ [5], OMPQ [6]), for each layer, given a bit-width, we use AdaRound [7] to quantize the layer's weights. Similar to previous works (GDFQ [8], Qimera [2], AdaDFQ [9], Genie [3], BRECQ [4], OMPQ [6], EMQ [5]), we use channel-wise quantizers for weights and layer-wise quantizers for activations. Regarding activation quantization, following the current state-of-the-art zero-shot/mixed-precision quantization works such as Genie [3], BRECQ [4], EMQ [5] and OMPQ [6], we use STE for optimizing the activation quantizer parameters.
>
>
> **Q2**:
> *The proposed method also works for conventional PTQ when the calibration data set is assumed not to fit the model statistics, which indicates that the method is not completely limited to ZSQ. Authors should provide more targeted explanations or provide experiments of the proposed method on PTQ for corroboration. Perhaps the proposed method can be used to build benchmarks on PTQ?*
>
> **Answer**:
> While the proposed method could be extended to conventional PTQ with a small set of real calibration data, we would like to re-emphasize that the focus of our paper is on zero-shot quantization (in which there is no access to any real original data) and mixed-precision quantization. Specifically, we propose a novel method that jointly optimizes both the synthesized calibration set and the bit-width of each layer in the context of zero-shot quantization. Therefore, an extensive comparison to post-training quantization methods, which require a set of real calibration data, is out of the scope of our paper. We would like to note that while we do not conduct an extensive comparison to PTQ methods, in Table 2 of the paper, we also compare our method (which is a zero-shot mixed-precision method) to some post-training mixed-precision quantization methods that require real data such as OMPQ [6], EMQ [5], FracBits-PACT [10], BRECQ [4]. The experimental results show that except for EMQ [5], our method outperforms other compared methods in all settings.
>
> **Q3**:
> *The traditional benchmark on CIFAR data set has not been adopted, and the authors are requested to add explanations.*
>
> **Answer**:
> We provide here results of the method on the CIFAR-10 and CIFAR-100 datasets. Specifically, we evaluate our method on these datasets using the W4A4 setting for the ResNet-18 model. The results are presented in the table below.
>
> | Method          | CIFAR-10 | CIFAR-100 |
> |-----------------|----------|-----------|
> | Qimera [2]         | 91.26    | 65.10     |
> | AdaSG [11]        | 92.10    | 66.42     |
> | IntraQ [12]        | 91.49    | 64.98     |
> | AdaDFQ [9]       | 92.31    | 66.81     |
> | Genie [3]          | 93.25    | 68.30     |
> | **Ours**        | **93.50**| **69.02** |

---

> ### Author Response · Authors · 2024-11-21
> **Response to Reviewer mxwe**
>
> ### Q4:
> *Line 235: “When that happens, we call the set X^(T) is k-equivalent to X^(V). We can optimize this objective, by matching the Gram-Gradient matrix of the two sets, according to Definition 1 and Theorem 3.1" ZSQ does not provide a validation set. I am trying to doubt the assumption in the article that when two samples are from synthetic datasets, their matching does not indicate the correctness of the feature.*
>
> **Answer**:
> To address the absence of a real validation set during model quantization, we synthesize a validation set, as detailed in lines 270 to 277 of the paper. Although the validation set is synthetic, it shares similar features with the original real data due to the warm-up process, which aligns the Batch Norm statistics with those of the full-precision model. Moreover, in Q2 of the General Response, we provide a visualization, demonstrating that using the synthetic validation set, our method can generate synthetic training samples with gradients that match the target gradients of the real validation set.
>
> **Q5**: *The generator configuration is not well aligned with the comparison method because Genie's generator is a newer version and this part needs further ablation.*
>
> **Answer**:
> We provide additional results when we use the generator from AdaDFQ [9] and GDFQ [8] methods in our framework.
> We evaluate our method with the W4A4 setting for the ResNet-18 model on the ImageNet dataset. As shown in the table below, our proposed framework, which incorporates the generator from the AdaDFQ [9] and GDFQ [8] methods, achieves significantly better performance compared to other methods. These results show that our method is applicable to different generators.
>
>
> **Table**: Comparisons of Top-1 classification accuracy (%) with state-of-the-art zero-shot quantization methods on the ImageNet dataset. Genie + MPQ denotes the combining of data generated using Genie [Genie] model with our bit allocation method. (*) indicates mixed-precision quantization. Methods with \\(\(\\diamond\)\\) use the same quantization setting as BRECQ [4] and Genie [3]. Our method maintains the same model size as other methods in the corresponding settings. FP denotes the full-precision model.
>
> | **Method**                     | **W/A** | **ResNet-18** |
> |--------------------------------|---------|---------------|
> | FP                             |         | 71.01         |
> | Qimera$^{\diamond}$ [2]           | 4/4     | 67.86         |
> |  IntraQ$^{\diamond}$ [12]            | 4/4     | 68.77         |
> | SADAG [13]                   | 4/4     | 69.72         |
> | Genie [3]              | 4/4     | 69.66         |
> | Genie + MPQ (Ours)         | */4     | 69.88         |
> |  Ours (AdaDFQ/GDFQ generator)  | */4     | 69.95         |
> | Ours (Genie generator)      | */4     | **70.05**     |
>
>
> **Q6**: *Lack of overhead analysis.*
>
> **Answer**:
> Regarding the overhead analysis, as mentioned in the paper, our method has a time complexity of $\mathcal{O}(|\mathbf{C}|L)$ for mixed-precision optimization, which is efficient since it scales linearly with the number of bit-width options and the number of layers in the model. In practice, we see that the mixed-precision optimization and zero-shot optimization processes have comparable overheads. Overall, experiments show that our framework incurs about 25% additional overhead compared to the Genie method [3].

---

> ### Author Response · Authors · 2024-11-21
> **Response to Reviewer mxwe**
>
> # References
>
> [1] Huantong Li, Xiangmiao Wu, Fanbing Lv, Daihai Liao, Thomas H. Li,
> Yonggang Zhang, Bo Han, and Mingkui Tan. Hard sample matters a lot
> in zero-shot quantization. In CVPR, 2023.
>
>
> [2] Kanghyun Choi, Deokki Hong, Noseong Park, Youngsok Kim, and Jinho
> Lee. Qimera: Data-free quantization with synthetic boundary supporting
> samples. In NeurIPS, 2021.
>
>
> [3] Yongkweon Jeon, Chungman Lee, and Ho-young Kim. Genie: Show me
> the data for quantization. In CVPR, 2023.
>
>
> [4] Yuhang Li, Ruihao Gong, Xu Tan, Yang Yang, Peng Hu, Qi Zhang, Fengwei
> Yu, Wei Wang, and Shi Gu. Brecq: Pushing the limit of post-training
> quantization by block reconstruction. ArXiv, abs/2102.05426, 2021.
>
>
> [5] Peijie Dong, Lujun Li, Zimian Wei, Xin-Yi Niu, Zhiliang Tian, and Hengyue
> Pan. EMQ: Evolving training-free proxies for automated mixed precision
> quantization. In ICCV, 2023.
>
>
> [6] Yuexiao Ma, Taisong Jin, Xiawu Zheng, Yan Wang, Huixia Li, Yongjian
> Wu, Guannan Jiang, Wei Zhang, and Rongrong Ji. OMPQ: Orthogonal
> mixed precision quantization. In AAAI, 2023.
>
>
> [7] Markus Nagel, Rana Ali Amjad, Mart Van Baalen, Christos Louizos, and
> Tijmen Blankevoort. Up or down? adaptive rounding for post-training
> quantization. In ICML, 2020.
>
>
> [8] Shoukai Xu, Shuhai Zhang, Jing Liu, Bohan Zhuang, Yaowei Wang, and
> Mingkui Tan. Generative data free model quantization with knowledge
> matching for classification. IEEE Transactions on Circuits and Systems
> for Video Technology, 33:7296–7309, 2023.
>
>
> [9] Biao Qian, Yang Wang, Richang Hong, and Meng Wang. Adaptive data-free quantization. In CVPR, 2023.
>
>
> [10] Jungwook Choi, Zhuo Wang, Swagath Venkataramani, Pierce I-Jen
> Chuang, Vijayalakshmi Srinivasan, and K. Gopalakrishnan. Pact: Parameterized clipping activation for quantized neural networks. ArXiv, 2018.
>
>
> [11] Biao Qian, Yang Wang, Richang Hong, and Meng Wang. Rethinking data-
> free quantization as a zero-sum game. In AAAI, 2023.
>
>
> [12] Yunshan Zhong, Mingbao Lin, Gongrui Nan, Jianzhuang Liu, Baochang
> Zhang, Yonghong Tian, and Rongrong Ji. Intraq: Learning synthetic images with intra-class heterogeneity for zero-shot network quantization. In
> CVPR, 2022.
>
>
> [13] Hoang Anh Dung, Cuong Pham, Trung Le, Jianfei Cai, and Thanh-Toan Do. Sharpness-aware data generation for zero-shot quantization. In ICML, 2024.

---

> ### Author Response · Authors · 2024-11-25
>
> Dear Reviewer mxwe,
>
> We hope the Reviewer has had time to look at our rebuttal. Could the Reviewer please share with us the Reviewer’s feedback on it?
>
> We sincerely appreciate the time and effort the Reviewer has dedicated to evaluating our paper and our response.

---

> > ### Comment · Reviewer_mxwe · 2024-11-28
> >
> > I admit that my concerns have not been fully addressed. In Table 1, AdaSG (Qian et al., 2023b) only quantized the weight, and used the naive quantization method of minmax and the naive STE fine-tuning method. However, Genie + MPQ (Ours) uses weight & activation quantization, and AdaRound [7] is used for weight quantization. I expect the authors to carefully list the quantizer baselines of the methods for fair comparison.

---

> > > ### Comment · Reviewer_mxwe · 2024-11-28
> > >
> > > I cannot observe the superiority of the proposed generated images from Table I because the quantizer configurations of the comparison methods are not well ablated.

---

> ### Author Response · Authors · 2024-12-01
> **Response to Reviewer mxwe**
>
> We sincerely thank the Reviewer for the constructive feedback.
> Firstly, we would like to summarize the contributions of our work, including:
> 1. We propose the first framework to jointly optimize both the data generation and the bit allocation for zero-shot mixed-precision settings.
> 2. We propose a novel approach for optimizing the calibration data based on the Gram-Gradient matrix of that calibration set.
> 3. We propose a scalable quadratic optimization approach that considers the impact of bit-width settings to the gradient of the model for bit allocation optimization.
>
> Secondly, we would like to note that while our method is applicable to different quantization schemes, in the paper, following previous works including Genie [3], BRECQ [4], EMQ [5] and OMPQ [6], we use AdaRound quantization scheme [7] for weight quantization and LSQ quantization scheme [14] for activation quantization.
> The experimental results show that our method significantly outperforms the current state-of-the-art zero-shot quantization methods.
>
> Thirdly, regarding AdaSG [11] and AdaDFQ [9], following the Reviewer's suggestion, we include here the results of our method after adopting the same weight and activation quantization scheme as AdaSG [11] and AdaDFQ [9], which use rounding-to-nearest and STE fine-tuning for weight quantization and the naive min-max method for activation quantization. The results for the ResNet-18 architecture are provided in Table below. Following the Reviewer's suggestion, we have carefully listed the quantizer baselines of the methods for fair comparison.
> As shown, our method achieves improvements of **1.92%** and **1.89%** for the 4/4 settings over AdaSG [11] and AdaDFQ [9], respectively.
>
> ### Table 1: Comparisons of Top-1 classification accuracy (%) with different zero-shot quantization methods on setting 4/4 of the ImageNet dataset on Resnet-18 architecture.
> *Genie + MPQ denotes the combining of generated data using Genie [3] model with our bit allocation method. Methods with $(\diamond)$ use the same quantization setting as Genie [3]. Methods with $(✦)$ use the same quantization scheme as AdaSG [11] and AdaDFQ [9].*
> | **Method**            	| **Acc**	|
> |---------------------------|-----------|
> | Qimera$^{✦}$ [2]      	| 63.84 	|
> | IntraQ$^{✦}$ [12]      	| 66.47 	|
> | AdaSG$^{✦}$ [11]      	| 66.50 	|
> | AdaDFQ$^{✦}$ [9]      	| 66.53 	|
> | Ours$^{✦}$            	| **68.42** |
> |---------------------------|----------|
> | Qimera$^{\diamond}$ [2]   | 67.86     |
> | IntraQ$^{\diamond}$ [12]   | 68.77     |
> | SADAG$^{\diamond}$ [13]                 | 69.72     |
> | Genie$^{\diamond}$ [3]                 | 69.66     |
> | Genie$^{\diamond}$ [3] + MPQ (Ours)    | 69.88     |
> | Ours$^{\diamond}$                      | **70.05** |
>
>
> We hope our response has addressed the Reviewer’s concerns. Please let us know if further details or elaboration are needed.
>
>
>
> [14] Steven K. Esser and Jeffrey L. McKinstry and Deepika Bablani and Rathinakumar Appuswamy and Dharmendra S. Modha. Learned Step Size Quantization. In ICLR, 2020.

---

> > ### Comment · Reviewer_mxwe · 2024-12-01
> > **follow up**
> >
> > The 4-bit case is difficult to judge because the task is relatively simple. The existing 3-bit quantization benchmarks still achieve poor accuracy, especially the AdaDFQ benchmarks (GDFQ series). Whether the authors are possible to show the experimental situation of 3bit？

---

> ### Author Response · Authors · 2024-12-01
> **Response to Reviewer mxwe**
>
> We sincerely thank the Reviewer for the follow-up feedback.
> We would like to note that the 4/4 setting is the most commonly adopted configuration in previous works. Following the Reviewer’s question, we provide the results of our method under the 3/3 setting after adopting the same weight and activation quantization schemes as AdaSG [11] and AdaDFQ [9]. The results for the ResNet-18 architecture are presented in Table below. As shown, using the same quantization scheme as AdaSG [11] and AdaDFQ [9], our method achieves significant improvements over AdaSG [11] and AdaDFQ [9] for the 3/3 settings.
>
>
>
>
>
>
>
>
> ### Table 1: Comparisons of Top-1 classification accuracy (%) with different zero-shot quantization methods on setting 3/3 of the ImageNet dataset on Resnet-18 architecture.
> *Genie + MPQ denotes the combining of generated data using Genie [3] model with our bit allocation method. Methods with $(\diamond)$ use the same quantization setting as Genie [3]. Methods with $(✦)$ use the same quantization scheme as AdaSG [11] and AdaDFQ [9].*
>
> | **Method**                | **Acc**    |
> |---------------------------|-----------|
> | IntraQ $^{✦}$ [12]          | -     |
> | Qimera $^{✦}$ [2]          | 1.17     |
> | AdaSG $^{✦}$ [11]          | 37.04     |
> | AdaDFQ $^{✦}$ [9]          | 38.10     |
> | Ours $^{✦}$                 | **55.65** |
> |---------------------------|----------|
> | SADAG$^{\diamond}$ [13]                 | 67.10     |
> | Genie$^{\diamond}$ [3]                 | 66.89     |
> | Genie$^{\diamond}$ [3] + MPQ (Ours)    | 67.33     |
> | Ours$^{\diamond}$                      | **67.57** |
>
> We hope our response has addressed the Reviewer’s concerns. Please let us know if further details or elaboration are needed.

---

> ### Comment · Reviewer_mxwe · 2024-12-02
> **Summary**
>
> Thank you for your rebuttal to my concerns. Most of my concerns are addressed, and I update my score.
>
> The idea proposed is innovative, however, the organization of this article still needs to be improved:
> 1. Analysis of synthetic images need to be decoupled from mixed-precision strategy to better understand the contribution of them.
> 2. Lack of comparison with state of the art in 2024, however I did observe superior performance in some configurations (especially W3A3 compared with TexQ, HAST [1-2]) when comparing them with SOTAs in 2023 and 2024 [1-3], which need to be added to the comparison table.
>
> I consider the revised article to be acceptable.
>
> [1] Xinrui Chen, Yizhi Wang, Renao Yan, Yiqing Liu, Tian Guan, and Yonghong He. Texq: Zero-shotnetwork quantization with texture feature distribution calibration. In NeurIPS, 2023.
>
> [2] Huantong Li, Xiangmiao Wu, Fanbing Lv, Daihai Liao, Thomas H Li, Yonggang Zhang, Bo Han,and Mingkui Tan. Hard sample matters a lot in zero-shot quantization. In CVPR, 2023a.
>
> [3] Chunxiao Fan, Ziqi Wang, Dan Guo, and Meng Wang. Data-free quantization via pseudo-labelfiltering. In CVPR, 2024.

---

> ### Author Response · Authors · 2024-12-02
> **Response to Reviewer mxwe**
>
> Dear Reviewer mxwe,
>
> We sincerely thank the Reviewer for increasing the score. We greatly appreciate the time and effort the Reviewer dedicated to considering our paper and our response. We will take the Reviewer's feedback into account in the revised paper.

---

### Official Review · Reviewer_Lh3R · 2024-10-30

**Soundness:** 3
**Presentation:** 2
**Contribution:** 2
**Rating:** 5
**Confidence:** 4

**Summary:**

This paper for the first time proposes a mechanism that combines zero-shot quantization and mixed-precision quantization.On the basis, the paper proposes a joint optimization framework between bit-width allocation and synthetic data generation. However, during the entire optimization process, these two parts are carried out alternately and iteratively. There is still a lack of a deeply coupled mechanism for joint optimization.

**Strengths:**

The inherent correlation between the quality of the generated calibration dataset and the bit allocation to the model's layers are considered. The paper is well-organized and clearly stated. I would suggest accepting it after the following concerns are addressed.

**Weaknesses:**

1. The references are all published until 2023. It would be best to provide some references on Mixed-precision Quantization/Zero-shot Quantization in 2024.
2. In Section 3.1, the authors mentioned that “In the realm of zero-shot quantization, the
validation set does not exist, so we only assume it here for explanation”. But the evaluation of model performance is conducted on the ImageNet dataset. Please carefully explain the relation between these two sets. If the validation does not exist, will there be significant changes or simplifications in formulas 6 and 8?
3. Please further explain the derivation process from formula 5 to formula 6.
4. In table 1 & table 2, the bit widths of results obtained using methods from this paper are marked as (*). From quantization setting in section 4.1, weights of some layers are set to 8 bits for initialization. Please calculate the average bit width of these models and fill it in the table, which will be more conducive to comparing the results and demonstrating the superiority of the method.
5. It would be better if the paper could compare the data synthesized in this paper with the data synthesized by existing methods and give deeper explanations or analyses.

**Questions:**

1. The references are all published until 2023. It would be best to provide some references on Mixed-precision Quantization/Zero-shot Quantization in 2024.
2. In Section 3.1, the authors mentioned that “In the realm of zero-shot quantization, the
validation set does not exist, so we only assume it here for explanation”. But the evaluation of model performance is conducted on the ImageNet dataset. Please carefully explain the relation between these two sets. If the validation does not exist, will there be significant changes or simplifications in formulas 6 and 8?
3. Please further explain the derivation process from formula 5 to formula 6.
4. In table 1 & table 2, the bit widths of results obtained using methods from this paper are marked as (*). From quantization setting in section 4.1, weights of some layers are set to 8 bits for initialization. Please calculate the average bit width of these models and fill it in the table, which will be more conducive to comparing the results and demonstrating the superiority of the method.
5. It would be better if the paper could compare the data synthesized in this paper with the data synthesized by existing methods and give deeper explanations or analyses.

---

> ### Author Response · Authors · 2024-11-21
> **Response to Reviewer Lh3R**
>
> ## **Response to Reviewer Lh3R**
>
> **We greatly appreciate the time and effort the Reviewer dedicated to considering our paper. Here are our responses to all concerns raised by the Reviewer.**
>
> ### **Weakness**
>
> **W1 and Q1**: *The references are all published until 2023. It would be best to provide some references on Mixed-precision Quantization/Zero-shot Quantization in 2024.*
>
> **Answer**:
> We sincerely thank the Reviewer for the suggestion. We have added a few new methods for mixed-precision quantization/zero-shot quantization in 2024 for comparison. Please refer to our General Response Q1.
>
> **W2 and Q2**:
> *In Section 3.1, the authors mentioned that “In the realm of zero-shot quantization, the validation set does not exist, so we only assume it here for explanation.” But the evaluation of model performance is conducted on the ImageNet dataset. Please carefully explain the relation between these two sets. If the validation does not exist, will there be significant changes or simplifications in formulas 6 and 8?*
>
> **Answer**:
> Please refer to our General Response Q2.
>
> **W3 and Q3**:
> *Please further explain the derivation process from formula 5 to formula 6.*
>
> **Answer**:
> We provide a detailed explanation from Equation 5 to Equation 6 in the General Response Q3.
>
> **W4 and Q4**:
> *In Table 1 and Table 2, the bit widths of results obtained using methods from this paper are marked as \(\*\). From quantization setting in Section 4.1, weights of some layers are set to 8 bits for initialization. Please calculate the average bit width of these models and fill it in the table, which will be more conducive to comparing the results and demonstrating the superiority of the method.*
>
> **Answer**:
> We sincerely thank the Reviewer for the suggestion. Please refer to our General Response Q1.
>
> **W5 and Q5**:
> *It would be better if the paper could compare the data synthesized in this paper with the data synthesized by existing methods and give deeper explanations or analyses.*
>
> **Answer**:
> We sincerely thank the Reviewer for the suggestion. In Table 1 of the paper, we have provided a comparison in performance between data synthesized by Genie and our data, i.e., the performance when combining the data generated by Genie [1] and our bit allocation method, compared with our framework.
> For deeper analysis of our synthetic data compared to the synthetic data generated by other methods, please refer to our General Response Q2.
>
> [1] Yongkweon Jeon, Chungman Lee, and Ho-young Kim. Genie: Show me the data for quantization. In CVPR, 2023.

---

> ### Author Response · Authors · 2024-11-25
>
> Dear Reviewer Lh3R,
>
> We hope the Reviewer has had time to look at our rebuttal. Could the Reviewer please share with us the Reviewer’s feedback on it?
>
> We sincerely appreciate the time and effort the Reviewer has dedicated to evaluating our paper and our response.

---

### Official Review · Reviewer_N47E · 2024-11-01

**Soundness:** 4
**Presentation:** 3
**Contribution:** 4
**Rating:** 6
**Confidence:** 3

**Summary:**

The paper presents a novel framework for zero-shot mixed-precision quantization (ZMPQ) that combines data generation and bit allocation optimization, focusing on improving quantization outcomes for deep learning models without access to original data. The approach introduces Gram-Gradient matrix-based data optimization and a scalable quadratic optimization for bit-width allocation, outperforming state-of-the-art methods on ImageNet benchmarks.

**Strengths:**

1.	Proposes a unique, joint optimization approach for data generation and bit allocation, filling a gap in zero-shot mixed-precision quantization research.
2.	Demonstrates superior or comparable performance to existing state-of-the-art methods under low-bit settings and varied model budgets, with results verified on multiple architectures (ResNet-18, ResNet-50, MobileNetV2).
3.	As a complex system, the paper offers a well-rounded suite of ablation studies that demonstrate the robustness of the method

**Weaknesses:**

I am not familiar with the work related to Mixed-precision quantization task. This paper is intuitive and reasonable on the whole. However there are some problems of clarity in the writing
1. Certain proofs, such as the explanation of Equation (6) and the construction and application of the Gram-Gradient matrix, are highly technical, demanding considerable background knowledge from the reader. The explanation provided for Equation (6), in particular, feels insufficient for those not deeply versed in the topic.
2. The paper would benefit from additional figures or qualitative results to aid in understanding complex concepts such as gradient matching, which currently lack sufficient illustration.
3. Table 1 and 2, could provide more detail on the W/A notation (e.g., clarifying 2/2 and */2) to enhance interpretability.

**Questions:**

Some of the suggestions for clarity mentioned above.

---

> ### Author Response · Authors · 2024-11-21
> **Response to Reviewer N47E**
>
> ## **Response to Reviewer N47E**
>
> **We greatly appreciate the time and effort the Reviewer dedicated to considering our paper. Here are our responses to all concerns raised by the Reviewer.**
>
> ### **Weakness**
>
> **W1**: *Certain proofs, such as the explanation of Equation (6) and the construction and application of the Gram-Gradient matrix, are highly technical, demanding considerable background knowledge from the reader. The explanation provided for Equation (6), in particular, feels insufficient for those not deeply versed in the topic.*
>
> **Answer**:
> We sincerely thank the Reviewer for the suggestion. We provide a more detailed explanation for Equation 5 and Equation 6 of our method. Please refer to our General Response Q3.
>
> **W2**: *The paper would benefit from additional figures or qualitative results to aid in understanding complex concepts such as gradient matching, which currently lack sufficient illustration.*
>
> **Answer**:
> We sincerely thank the Reviewer for the suggestion. The gradient matching score \\( {\mathcal{J}\_{V,i}\^{\(\theta_Q\)}}\^T \\mathcal{J}_{T,j}\^{(\\theta\_Q)} \\) reflects how effectively the model can minimize the loss on the validation sample \\( x\^{\(V\)}\_i \\) indirectly by training on the sample \\( x\^{\(T\)}\_{j} \\). When the gradient matching scores between training samples from \\( X\^{\(T\)} \\) and validation samples \\( X\^{\(V\)} \\) are high, the model quantized over the training set \\( X\^{\(T\)} \\) can minimize the loss over the validation set \\( X\^{\(V\)} \\). Please refer to our General Response Q2 for the visualization of gradient matching scores between the real validation set and our generated training set, compared to that of other zero-shot data generation methods.
>
> **W3**: *Table 1 and 2, could provide more detail on the W/A notation (e.g., clarifying 2/2 and \*/2) to enhance interpretability.*
>
> **Answer**:
> We sincerely thank the Reviewer for the suggestion. W/A denotes the bit-width configuration for weights/activations of the model (e.g., 2/2 means both weights and activations are quantized to 2 bits). The \\( (*) \\) symbol denotes mixed-precision quantization (e.g., \\( */2 \\) means the activations of the model are quantized to 2 bits, while mixed-precision quantization is used for the model weights, under the constraint that their total bits usage does not exceed that of uniform 2-bit quantization for the model weights). Please refer to our General Response Q1.

---

> > ### Comment · Reviewer_N47E · 2024-11-25
> >
> > I appreciate the clarifications provided. However, as I am not deeply familiar with the related works in this area, it is difficult for me to fully evaluate the significance of this contribution. I will keep my original score but raise my confidence level.

---

> ### Author Response · Authors · 2024-12-04
> **Response to Reviewer N47E**
>
> Dear Reviewer N47E,
>
> We sincerely thank the Reviewer for the time and effort the Reviewer dedicated to considering our paper and our response.

---

### Author Response · Authors · 2024-11-21
**General Response**

# References

[1] Hoang Anh Dung, Cuong Pham, Trung Le, Jianfei Cai, and Thanh-Toan Do. Sharpness-aware data generation for zero-shot quantization. In ICML, 2024.

[2] Ke Xu, Zhongcheng Li, Shanshan Wang, and Xingyi Zhang. Ptmq: Post- training multi-bit quantization of neural networks. In AAAI, 2024.

[3] Yongkweon Jeon, Chungman Lee, and Ho-young Kim. Genie: Show me the data for quantization. In CVPR, 2023.

[4] Yuhang Li, Ruihao Gong, Xu Tan, Yang Yang, Peng Hu, Qi Zhang, Fengwei Yu, Wei Wang, and Shi Gu. Brecq: Pushing the limit of post-training quantization by block reconstruction. ArXiv, abs/2102.05426, 2021.

[5] Yunshan Zhong, Mingbao Lin, Gongrui Nan, Jianzhuang Liu, Baochang Zhang, Yonghong Tian, and Rongrong Ji. Intraq: Learning synthetic im- ages with intra-class heterogeneity for zero-shot network quantization. In CVPR, 2022.

[6] Kanghyun Choi, Deokki Hong, Noseong Park, Youngsok Kim, and Jinho Lee. Qimera: Data-free quantization with synthetic boundary supporting samples. NeurIPS, 2021.

[7] Biao Qian, Yang Wang, Richang Hong, and Meng Wang. Rethinking data- free quantization as a zero-sum game. AAAI, 2023.

[8] Biao Qian, Yang Wang, Richang Hong, and Meng Wang. Adaptive data- free quantization. CVPR, 2023.

[9] Jungwook Choi, Zhuo Wang, Swagath Venkataramani, Pierce I-Jen Chuang, Vijayalakshmi Srinivasan, and K. Gopalakrishnan. Pact: Param- eterized clipping activation for quantized neural networks. ArXiv, 2018.

[10] Yuexiao Ma, Taisong Jin, Xiawu Zheng, Yan Wang, Huixia Li, Yongjian Wu, Guannan Jiang, Wei Zhang, and Rongrong Ji. OMPQ: Orthogonal mixed precision quantization. In AAAI, 2023.

[11] Peijie Dong, Lujun Li, Zimian Wei, Xin-Yi Niu, Zhiliang Tian, and Hengyue Pan. EMQ: Evolving training-free proxies for automated mixed precision quantization. ICCV, 2023.

[12] Yaohui Cai, Zhewei Yao, Zhen Dong, Amir Gholami, Michael W. Mahoney, and Kurt Keutzer. ZeroQ: A novel zero shot quantization framework. CVPR, 2020.

[13] Zhewei Yao, Zhen Dong, Zhangcheng Zheng, Amir Gholami, Jiali Yu, Eric Tan, Leyuan Wang, Qijing Huang, Yida Wang, Michael W. Mahoney, and Kurt Keutzer. HAWQV3: Dyadic neural network quantization. ArXiv, 2020.

[14] Huantong Li, Xiangmiao Wu, Fanbing Lv, Daihai Liao, Thomas H. Li, Yonggang Zhang, Bo Han, and Mingkui Tan. Hard sample matters a lot in zero-shot quantization. In CVPR, 2023.

---

### Author Response · Authors · 2024-11-21
**General Response**

**Q3**: *More detailed explanation for Equations 5 and 6*

**Answer**:
Our optimization objective for the generated dataset is:

\\begin{aligned}[b]
     & X^{\(T\)} = \\arg \\min\_{ X^{\(T\)}}
     \\mathcal{L}\_{R}\( \\theta\_{Q}^{\*},\\theta\_{FP},X^{\(V\)},B\) \\\\
     &\\text{s.t.:\\ } \\theta\_{Q}^{\*} = \\arg \\min\_{\\theta\_Q} \\mathcal{L}\_{R}\( \\theta\_{Q},\\theta\_{FP},X^{\(T\)},B\),
\\end{aligned}


where \\\( \\theta\_Q^{\*} \\\)  is the model weights after updating \\\( \\theta\_Q \\\) with \\\( X^{\(T\)} \\\) under the bit-width setting \\\( B \\\).
Define \\\( \\delta\_{\\theta\_{Q}} = \\theta\_{Q}^{\*}-\\theta\_{Q} \\\) as the weight difference of the quantized model \\\( \\theta\_{Q} \\\) before and after quantization with the training dataset \\\( X^{\(T\)} \\\). In practice, we only approximate the calibrated model using a single-step gradient descent, which yields:

\\\(
\\delta\_\{\\theta\_{Q}} = -\\alpha \\nabla\_{\\theta\_Q} \\mathcal{L}\_{R}\(\\theta\_{Q},\\theta\_{FP},X^{\(T\)},B\), \\\)


where \\\( \\alpha \\\) denotes the learning rate. Using the first-order Taylor expansion for the reconstruction loss \\\( \\mathcal{L}\_{R}\(\\theta\_{Q}^{\*},\\theta\_{FP},X^{\(T\)},B\) \\\) at \\\( \\theta\_{Q} \\\), we have:

\\\[
\\begin{aligned}[b]
     \\arg \\min\_{ X^{\(T\)}}\\mathcal{L}\_{R}\(\\theta^{\*}\_{Q},\\theta\_{FP},X^{\(V\)},B\)
     &= \\arg \\min\_{ X^{\(T\)}} \\mathcal{L}\_{R}\(\\theta\_{Q},\\theta\_{FP},X^{\(V\)},B\)  + \\nabla\_{\\theta\_Q} \\mathcal{L}\_{R}\(\\theta\_{Q},\\theta\_{FP},X^{\(V\)},B\)^T \(\\theta^{\*}\_{Q} - \\theta\_Q\).
\\end{aligned}
\\\]

The first term \\\( \\mathcal{L}\_{R}\(\\theta\_{Q},\\theta\_{FP},X^{\(V\)},B\) \\\) is independent of \\\( X^{\(T\)} \\\), so we can ignore it:

\\\[
\\begin{aligned}[b]
     \\arg \\min\_{ X^{\(T\)}}\\mathcal{L}\_{R}\(\\theta^{\*}\_{Q},\\theta\_{FP},X^{\(V\)},B\)
     = \\arg \\min\_{ X^{\(T\)}} \\nabla\_{\\theta\_Q} \\mathcal{L}\_{R}\(\\theta\_{Q},\\theta\_{FP},X^{\(V\)},B\)^T \(\\theta^{\*}\_{Q} - \\theta\_Q\).
\\end{aligned}
\\\]

If we replace \\\( \(\\theta^{\*}\_{Q} - \\theta\_Q\) = \\delta\_{\\theta\_{Q}} = -\\alpha \\nabla\_{\\theta\_Q} \\mathcal{L}\_{R}\(\\theta\_{Q},\\theta\_{FP},X^{\(T\)},B\) \\\), we have:

\\\[
\\begin{aligned}[b]
     \\arg \\min\_{ X^{\(T\)}}\\mathcal{L}\_{R}\(\\theta^{\*}\_{Q},\\theta\_{FP},X^{\(V\)},B\)
     &= \\arg \\min\_{ X^{\(T\)}} \\nabla\_{\\theta\_Q} \\mathcal{L}\_{R}\(\\theta\_{Q},\\theta\_{FP},X^{\(V\)},B\)^T \\delta\_{\\theta\_{Q}} \\\\
     &= \\arg \\min\_{ X^{\(T\)}} -\\alpha \\nabla\_{\\theta\_Q} \\mathcal{L}\_{R}\(\\theta\_{Q},\\theta\_{FP},X^{\(V\)},B\)^T \\nabla\_{\\theta\_Q} \\mathcal{L}\_{R}\(\\theta\_{Q},\\theta\_{FP},X^{\(T\)},B\).
\\end{aligned}
\\\]

We denote the gradient vector of the model evaluated on the training set as \\\( \\mathcal{J}\_{T}^{\(\\theta\_Q\)} \\\):

\\\[
\\begin{aligned}[b]
     \\mathcal{J}\_{T}^{\(\\theta\_Q\)} = \\nabla\_{\\theta\_Q} \\mathcal{L}\_{R}\(\\theta\_{Q},\\theta\_{FP},X^{\(T\)},B\) = \\frac{1}{|X^{\(T\)}|} \\sum\_{j=1}^{|X^{\(T\)}|} \\mathcal{J}\_{T,j}^{\(\\theta\_Q\)},
\\end{aligned}
\\\]

where \\\( \\mathcal{J}\_{T,j}^{\(\\theta\_Q\)} \\\) denotes the gradient vector of the reconstruction loss evaluated on the \\\( j^{\\text{th}} \\\) sample in the training set \\\( X^{\(T\)} \\\) w.r.t. the model weight \\\( \\theta\_Q \\\). Similarly, we have:

\\\[
\\begin{aligned}[b]
     \\mathcal{J}\_{V}^{\(\\theta\_Q\)} = \\nabla\_{\\theta\_Q} \\mathcal{L}\_{R}\(\\theta\_{Q},\\theta\_{FP},X^{\(V\)},B\) = \\frac{1}{|X^{\(V\)}|} \\sum\_{i=1}^{|X^{\(V\)}|} \\mathcal{J}\_{V,i}^{\(\\theta\_Q\)}.
\\end{aligned}
\\\]

Now, the optimization objective becomes:

\\\[
\\begin{aligned}[b]
     \\arg \\min\_{ X\^{\(T\)}} \\mathcal{L}\_{R}\( \\theta\_{Q}^{\*},\\theta\_{FP},X\^{\(V\)}, B\)
     &= \\arg \\min\_{ X\^{\(T\)}} -\\alpha {\\mathcal{J}\_{V}\^{\(\\theta\_Q\)}}\^T \\mathcal{J}\_{T}\^{\(\\theta\_Q\)} \\\\
     &= \\arg \\max\_{X^{\(T\)}} \\frac{1}{|X^{\(V\)}|} \\sum\_{i=1}^{|X^{\(V\)}|} \\frac{1}{|X^{\(T\)}|} \\sum\_{j=1}^{|X^{\(T\)}|} {\\mathcal{J}\_{V,i}^{\(\\theta\_Q\)}}^T \\mathcal{J}\_{T,j}^{\(\\theta\_Q\)} \\\\
     &= \\arg \\max\_{X^{\(T\)}} \\frac{1}{|X^{\(V\)}|} \\sum\_{i=1}^{|X^{\(V\)}|} \\frac{1}{|X^{\(T\)}|} \\sum\_{j=1}^{|X^{\(T\)}|} I\(x^{\(V\)}\_i, x^{\(T\)}\_j\),
\\end{aligned}
\\\]

where \\\( I\(x^{\(V\)}\_i, x^{\(T\)}\_j\) = \{\mathcal{J}\_{V,i}^{\(\\theta\_Q\)}}^T \\mathcal{J}\_{T,j}^{\(\\theta\_Q\)} \\\) is a gradient matching score, denoting how well the model can minimize the loss on sample \\\( x^{\(V\)}\_i \\\) indirectly through training sample \\\( x^{\(T\)}\_j \\\). Therefore, we want the gradient matching score of all validation samples \\\( x^{\(V\)}\_i \\\) to remain high throughout the entire training process.

---

### Author Response · Authors · 2024-11-21
**General Response**

Q2: Regarding the validation set and visualization.

Answer:
In the context of zero-shot quantization, we have access to the weight of the pretrained model, but do not have access to any real original images during the quantization process. In the derivation of our method, we assume the availability of the validation set in order to derive the optimization objective. In practice, when quantizing the model, we need to circumvent the lack of the real validation set. To this end, we synthesize the validation set as explained from line 270 to line 277 of the paper. Before the optimization process, we warm-up the synthetic training data, by matching its Batch Norm statistics with those of the full-precision model. The synthetic validation set is then generated by sampling data in the neighborhood of the warmed-up training samples. By matching the Batch Norm statistics in the warm-up stage, the validation samples are likely to share features with the real data.
After the model has been quantized, we evaluate the quantized model on the real validation set of the ImageNet dataset to compare to other methods. As we mentioned above, the validation set is synthesized in our framework. Hence, there are no changes to Eqs. (6) and (8).

As described from line 225 to line 231 of the paper, we want to encourage each validation sample to have a high gradient matching score with at least one training sample. To demonstrate the effectiveness of our method, we visualize the distribution of gradient matching scores between the real validation set and our generated training set, compared to that of other zero-shot data generation methods [3,14]. Specifically, for each real validation sample, we compute its maximum gradient matching score with a synthetic sample from the training set generated by each method and plot the score distribution for the entire validation set.
The visualization can be accessed via the following link: [Visualization of Gradient Similarity](https://anonymous.4open.science/r/test_repo-D26B/Visualization_Gradient_Similarity_9.pdf).

The results show that our Gram-Gradient matching approach leads to better gradient matching over the real validation data, compared to other zero-shot quantization methods. This demonstrates that Gram-Gradient matrix matching between the synthetic training set and the synthetic validation set can lead to a better gradient matching between the synthetic training set and the real validation data.

---

### Author Response · Authors · 2024-11-21
**General Response**

# General Response

**We sincerely thank the Reviewers for the constructive feedback.**

## Question

**Q1**: *Regarding Table 1 and Table 2 of the paper.*

**Answer**:
Following the feedbacks from the Reviewers, we have updated Tables 1 and 2 below, including adding the average bit, clarification for notations and additional comparison methods. We provide additional methods published in 2024 for zero-shot quantization \(SADAG [1]\) and mixed-precision quantization \(PTMQ [2]\) for comparison in Tables 1 and 2 below.
It is worth noting that, to the best of our knowledge, there is no available joint optimization method for both mixed-precision quantization and zero-shot quantization published in 2024.

### Table 1: Comparisons of Top-1 classification accuracy (%) with the state-of-the-art zero-shot quantization methods on the ImageNet dataset.

*Genie + MPQ denotes the combining of generated data using Genie[3] model with our bit allocation method. Results marked with $(\dagger)$ are reproduced using the official released code of the corresponding paper. $(\*)$ indicates mixed-precision quantization for models' weights. Methods with $(\diamond)$ use the same quantization setting as BRECQ [4] and Genie [3]. **AverageBit** denotes the average bit of models in the same block, accounting for layers configured to 8 bits. Our method maintains the same model size as other methods in the corresponding settings. FP denotes the full-precision model.*

| **Method**                | **W/A**  | **Resnet-18**             |   | **Resnet-50**             |   | **MobilenetV2**            |   |
|---------------------------|----------|---------------------------|---|---------------------------|---|---------------------------|---|
|                           |          | **AverageBit**   | **Acc**    | **AverageBit**   | **Acc**    | **AverageBit**   | **Acc**    |
|---------------------------|----------|-----------|-----------|-----------|-----------|-----------|-----------|
| FP                        |          | 32/32      | 71.01     |  32/32          | 76.63     |    32/32        | 72.20     |
| SADAG [1]         | 2/2      |           | 54.51     |           | 57.55     |           | 13.01     |
| Genie [3]        | 2/2      |           | 53.74     |           | 56.81     |           | 11.93     |
| Genie [3] + MPQ (Ours)        | */2      | 2.2/2     | 58.33     | 2.48/2    | 63.85     | 4.2/2     | 23.39     |
| Ours                      | */2      |           | **58.60** |           | **64.23** |           | **23.69** |
|---------------------------|----------|-----------|-----------|-----------|-----------|-----------|-----------|
| IntraQ [5]      | 3/3      |           | -         |           | -         |           | -         |
| Qimera [6]      | 3/3      | 3/3       | 1.17      | 3/3       |           | 3/3       | -         |
| AdaSG [7]        | 3/3      |           | 37.04     |           | 16.98     |           | 26.90     |
| AdaDFQ [8]        | 3/3      |           | 38.10     |           | 17.63     |           | 28.99     |
|---------------------------|----------|-----------|-----------|-----------|-----------|-----------|-----------|
| Genie [3]        | 3/3      |           | 66.89     |           | 72.54     |           | $55.13^\dagger$ |
| Genie [3] + MPQ (Ours)        | */3      | 3.2/3     | 67.33     | 3.4/3     | 73.75     | 4.8/3     | 62.45     |
| SADAG [1]        | 3/3      |           | 67.10     |           | 72.62     |           | 56.02     |
| Ours                      | */3      |           | **67.57** |           | **73.83** |           | **62.60** |
|---------------------------|----------|-----------|-----------|-----------|-----------|-----------|-----------|
| Qimera [6]      | 4/4      |           | 63.84     |           | 66.25     |           | 61.62     |
| AdaSG [7]        | 4/4      | 4/4       | 66.50     | 4/4       | 68.58     | 4/4       | 65.15     |
| IntraQ [5]      | 4/4      |           | 66.47     |           | -         |           | 65.10     |
| AdaDFQ [8]          | 4/4      |           | 66.53     |           | 68.38     |           | 65.41         |
|---------------------------|----------|-----------|-----------|-----------|-----------|-----------|-----------|
| Qimera$^{\diamond}$ [6] | 4/4      |           | 67.86     |           | 72.90     |           | 58.33     |
| IntraQ$^{\diamond}$ [5] | 4/4      |           | 68.77     |           | 68.16     |           | 63.78     |
| SADAG [1]        | 4/4      |           | 69.72     |           | 75.7      |           | 68.54     |
| Genie [3]        | 4/4      |  4.17/4         | 69.66     |   4.3/4        | $75.57^\dagger$ |  5.47/4      | 68.38     |
| Genie [3] + MPQ (Ours)        | */4      |           | 69.88     |           | 75.84     |           | 69.95     |
| Ours                      | */4      |           | **70.05** |           | **75.87** |           | **70.07** |

---

> ### Author Response · Authors · 2024-11-21
> **General Response**
>
> ### Table 2: The comparisons of our proposed method with the state-of-the-art mixed-precision quantization methods under different model size budgets on ResNet-18.
> *$(\*)$ indicates mixed-precision quantization for models' weights or activations. **AverageBit** denotes the average bit of the model, accounting for layers configured to 8 bits. The last column **Zero-shot** denotes whether the framework requires real data.*
>
> | **Method Name**                | **W/A** | **Model Size (MB)** | **AverageBit**   | **Top-1 (%)** | **Zero-shot** |
> |---------------------------------|---------|---------------------|----------|---------------|---------------|
> | FracBits-PACT [9] | \*/\*     | 4.5                 | 3.23/4   | 69.10         | -             |
> | **OMPQ** [10]            | */4     | 4.5                 | 3.23/4   | 68.89         | $\times$      |
> | **EMQ** [11]              | */4     | 4.5                 | 3.23/4   | **69.66**     | $\times$      |
> | **Ours**                        | */4     | 4.5                 | 3.23/4   | 69.38         | ✔️    |
> | **ZeroQ** [12]          | 4/4     | 5.81                | 4.17/4   | 21.20         | ✔️    |
> | **BRECQ** [4]  | 4/4     | 5.81                | 4.17/4   | 69.32         | $\times$      |
> | **PACT** [9]  | 4/4     | 5.81                | 4.17/4   | 69.20         | $\times$      |
> | **HAWQ-V3** [13] | 4/4   | 5.81                | 4.17/4   | 68.45         | $\times$      |
> | **FracBits-PACT** [9] | \*/\* | 5.81               | 4.17/4   | 69.70         | $\times$      |
> | **PTMQ** [2]            | */4     | 5.5                 | 3.95/4   | 67.57         | $\times$      |
> | **OMPQ** [10]            | */4     | 5.5                 | 3.95/4   | 69.38         | $\times$      |
> | **EMQ** [11]              | */4     | 5.5                 | 3.95/4   | **70.12**     | $\times$      |
> | **Ours**                        | */4     | 5.81                | 4.17/4   | 70.06         | ✔️   |
> | **BRECQ** [4]  | */8     | 4.0                 | 2.87/8   | 68.82         | $\times$      |
> | **OMPQ** [10]            | */8     | 4.0                 | 2.87/8   | 69.41         | $\times$      |
> | **EMQ** [11]              | */8     | 4.0                 | 2.87/8   | **69.92**     | $\times$      |
> | **Ours**                        | */8     | 4.0                 | 2.87/8   | 69.86         | ✔️    |

---

### Author Response · Authors · 2024-11-21
**Revision**

We sincerely thank the Reviewers for their thoughtful feedback and valuable comments on our work. We have slightly revised the paper to address typos.

---

### Meta-Review · Area_Chair_5Nyz · 2024-12-21

**Metareview:**

This paper proposes a joint optimization approach for both the calibration set and the bit-width of each layer within the context of zero-shot quantization. The scores for this paper are mixed, with one reviewer rating it marginally above acceptance but with low confidence, one rating it marginally below acceptance, and one recommending acceptance. After carefully reviewing the feedback and discussions between the reviewers and authors, the Area Chair (AC) has concluded that the paper cannot be accepted. The approach, which combines joint optimization of calibration data and bit allocation across model layers, lacks novelty. Additionally, the paper fails to compare its methods with recent approaches from 2024, and in some cases, the proposed method performs worse than existing ones under certain configurations. Finally, the overall writing and organization of the paper make it difficult for readers to follow and understand.

**Additional Comments On Reviewer Discussion:**

This paper received mixed scores. After checking the comments and discussions, the AC assumes there exist drawbacks for the current version. As such, this paper cannot be accepted in the current version.

---

### Decision · Program_Chairs · 2025-01-22

Reject